# A Model to Predict Upstaging to Invasive Carcinoma in Patients Preoperatively Diagnosed with Low-Grade Ductal Carcinoma In Situ of the Breast

**DOI:** 10.3390/cancers14020370

**Published:** 2022-01-12

**Authors:** Luca Nicosia, Anna Carla Bozzini, Silvia Penco, Chiara Trentin, Maria Pizzamiglio, Matteo Lazzeroni, Germana Lissidini, Paolo Veronesi, Gabriel Farante, Samuele Frassoni, Vincenzo Bagnardi, Cristiana Fodor, Nicola Fusco, Elham Sajjadi, Enrico Cassano, Filippo Pesapane

**Affiliations:** 1Breast Imaging Division, Radiology Department, IEO, European Institute of Oncology, IRCCS, 20141 Milan, Italy; anna.bozzini@ieo.it (A.C.B.); silvia.penco@ieo.it (S.P.); chiara.trentin@ieo.it (C.T.); maria.pizzamiglio@ieo.it (M.P.); enrico.cassano@ieo.it (E.C.); filippo.pesapane@ieo.it (F.P.); 2Division of Cancer Prevention and Genetics, IEO, European Institute of Oncology, IRCCS, 20141 Milan, Italy; matteo.lazzeroni@ieo.it; 3Division of Breast Surgery, IEO, European Institute of Oncology, IRCCS, 20141 Milan, Italy; germana.lissidini@ieo.it (G.L.); paolo.veronesi@ieo.it (P.V.); gabriel.farante@ieo.it (G.F.); 4Department of Oncology and Hemato-Oncology, University of Milan, 20133 Milan, Italy; nicola.fusco@ieo.it (N.F.); elham.sajjadi@ieo.it (E.S.); 5Department of Statistics and Quantitative Methods, University of Milan-Bicocca, 20126 Milan, Italy; samuele.frassoni@unimib.it (S.F.); vincenzo.bagnardi@ieo.it (V.B.); 6Division of Radiation Oncology, IEO, European Institute of Oncology, IRCCS, 20141 Milan, Italy; cristiana.fodor@ieo.it; 7Division of Pathology, IEO, European Institute of Oncology, IRCSS, 20141 Milan, Italy

**Keywords:** ductal carcinoma in situ (DCIS), invasive breast carcinoma, breast, biopsy, overtreatment, active surveillance

## Abstract

**Simple Summary:**

Surgical management is currently the main standard of care procedure used in order to treat ductal carcinoma in situ (DCIS) of the breast. Nevertheless, the survival benefit of surgical resection in patients with such lesions appears to be low, especially for low-grade DCIS. Low-grade DCIS typically exhibit a slow growth pattern and, in many cases, never fully develop into a clinically significant disease: discerning harmless lesions from potentially invasive ones could lead to avoid overtreatment in many patients. Nonetheless, up to 26% of patients with biopsy-proven DCIS can reveal a synchronous invasive carcinoma in surgical specimens. Here, we aimed to create a model of radiological and pathological criteria able to reduce the underestimation of vacuum assisted breast biopsy in DCIS, identifying patients at very low risk (e.g., <2%) of diagnostic underestimation.

**Abstract:**

Background: We aimed to create a model of radiological and pathological criteria able to predict the upgrade rate of low-grade ductal carcinoma in situ (DCIS) to invasive carcinoma, in patients undergoing vacuum-assisted breast biopsy (VABB) and subsequent surgical excision. Methods: A total of 3100 VABBs were retrospectively reviewed, among which we reported 295 low-grade DCIS who subsequently underwent surgery. The association between patients’ features and the upgrade rate to invasive breast cancer (IBC) was evaluated by univariate and multivariate analysis. Finally, we developed a nomogram for predicting the upstage at surgery, according to the multivariate logistic regression model. Results: The overall upgrade rate to invasive carcinoma was 10.8%. At univariate analysis, the risk of upgrade was significantly lower in patients with greater age (*p* = 0.018), without post-biopsy residual lesion (*p* < 0.001), with a smaller post-biopsy residual lesion size (*p* < 0.001), and in the presence of low-grade DCIS only in specimens with microcalcifications (*p* = 0.002). According to the final multivariable model, the predicted probability of upstage at surgery was lower than 2% in 58 patients; among these 58 patients, only one (1.7%) upstage was observed, showing a good calibration of the model. Conclusions: An easy-to-use nomogram for predicting the upstage at surgery based on radiological and pathological criteria is able to identify patients with low-grade carcinoma in situ with low risk of upstaging to infiltrating carcinomas.

## 1. Introduction

Breast cancer is one of the most prevalent malignancies among women worldwide, still leading to a considerable incidence of death; in 2020, almost 685,000 women were deceased owing to this malignancy [1]. Ductal carcinoma in situ (DCIS) represents almost 20–25% of all breast neoplastic lesions being diagnosed [2,3]. In DCIS, the cancer cells’ growth is confined to the breast ducts or lobules with a minimal potential to spread [4]. As DCIS is mainly clinically occult (around 9% are symptomatic), more than 90% of cases are detected only through imaging studies. Prior to 1980, this condition could be rarely identified. Owing to the improvement of diagnostic and screening imaging tools, specifically mammography, DCIS incidence has rapidly increased [5]. 

According to the Current National Comprehensive Cancer Network (NCCN), the best therapeutic options are recommended as mastectomy, lumpectomy with radiation, or lumpectomy alone with the potential addition of tamoxifen for hormone receptor–positive carcinoma in situ [6]. There are few data available that compare the benefit obtained from the currently recommended treatments with those who did not receive treatment (active surveillance) [7]. 

Carcinoma in situ of the breast does not present a risk of invasion and metastasis and the mortality rate is as low as 4% [7]. Therefore, the main purpose of the treatment is to prevent the development of invasive carcinoma. However, a meta-analysis of underestimation and predictors of invasive breast cancer showed that up to 26% of patients with biopsy-proven DCIS can reveal a synchronous invasive carcinoma in surgical specimens [8]. As this percentage is unacceptable, it is necessary to reduce the diagnostic underestimation of the VABB before proposing active surveillance to patients.

How many low-grade breast carcinomas in situ are actually infiltrating carcinomas or high-grade carcinomas in situ? How can we identify patients at low risk of being underestimated with the VABB?

In our study, we examined surgical specimens of patients diagnosed with low-grade DCIS to identify potential indicators for upgrading [9]. 

By selecting a population with a low risk of upgrading, we may identify patients with low-grade breast cancer in which surgery may be safely spared.

Four prospective international study protocols (LORIS, COMET, LORD, and LORETTA) are currently in place to evaluate non-invasive treatment strategies for DCIS; however, a selection of patient population based on clinical and radiological features (which may reduce the diagnostic underestimation of the biopsy) appears to be neglected in these protocols [10]. We firmly believe that a better initial selection of patients, based on radiological, pathological, and clinical features, can make these protocols more effective, reducing the diagnostic underestimation of the biopsy. We can, therefore, hypothesize models that identify a population in which active surveillance could be safer. 

The Loretta protocol is the only one that takes into account the initial size of the lesion. However, other radiological and pathological features, which would be easy to use to reduce the diagnostic underestimation of the biopsy, are not considered in the initial selection of patients in the four prospective international study protocols.

Details are shown in Table 1.

The purpose of our study is to identify a predictive model that identifies the features, mainly based on imaging, that can predict the diagnostic underestimation of low-grade DCIS to invasive carcinoma or worst grade DCIS. 

## 2. Materials and Methods

This retrospective study was notified to the Ethics Committee (Identification Number UID 2897, 24 September 2021) and was approved by the Institutional Review Board.

### 2.1. Study Design and Population

We retrospectively studied all patients who underwent a screening mammogram or an ultrasound for prevention, i.e., dense breasts in a single referral canter for breast cancer care (European Institute of Oncology, Milan, Italy). Among which those with doubtful lesions, between January 1999 and January 2019, were included in our study cohort. All the lesions were classified according to the Breast Imaging Reporting and Data System (BI-RADS), using the score BIRADS ≥ 3 as a threshold to define suspicious lesions. Ultrasound- or stereotactic-guided VABB was performed in patients with BIRADS ≥ 4; only in exceptional cases (3/295), with very high familiarity for breast cancer, patients with BIRADS 3 were biopsied too [15,16,17].

All the lesions undergoing stereotactic VABB presented as microcalcifications. Before each stereotactic VABB, two projection mammograms were performed in order to assess the precise extension of the lesion (Figure 1).

After the VABB procedure, all patients underwent two additional mammogram projections to confirm the complete macroscopic removal or the presence of residual lesion.

Before each ultrasound VABB, both transverse and longitudinal static images were acquired by US performed prior to the biopsy. After the procedure both transverse and longitudinal US images were taken to detect the complete macroscopic removal or the presence of residual of the lesion in all patients.

We collected and retrospectively analysed some of the features reported in the radiologist’s and pathologist’s report, in particular: the number of cores obtained for each biopsy, the complete macroscopical removal of the lesion, the diameter of the biopsy needle, and—for stereotactic VABB—if the disease was present only in the cores with microcalcifications (or even in the cores without microcalcifications, if any).

We investigated a potential correlation between patient’s age, lesion size, diameter of the needle (with an equal number of biopsy samples, more tissue is collected with a larger needle), number of cores, complete macroscopic removal of the lesion, cases showing low-grade DCIS only in cores with microcalcifications, and the chance of upgrade to a worst grade DCIS or invasive ductal carcinoma (IDC). Since the BIRADS is often very subjective [18], we have excluded it from the analysis. Figure 2 represents a low-grade DCIS.

In accordance with some recent studies that have shown a benefit in the change of therapy with patients presenting an intermediate grade DCIS with Ki-67 > 14%, we considered this threshold to be significant in our underestimation analyses of worst grade carcinoma in situ at the biopsy [19]. In our predictive model, we also considered underestimation of carcinoma in situ of the worst degree in case of finding of intermediate grade DCIS with Ki-67 > 14%.

### 2.2. Statistical Analysis

Continuous data are reported as median and range and categorical data are reported as counts and percentages.

Univariate and multivariate logistic regressions were performed to assess the association of age, biopsy needle, residual disease, residual lesion size, number of cores and disease only in cores with micro, with the risk of upgrade of low-grade DCIS to invasive carcinoma. Age, residual lesion size and number of cores were considered as continuous predictors in the models. Residual lesion size was log-transformed due to its skewed distribution.

According to the multivariate logistic regression model, a nomogram for predicting the upstage at surgery was reported.

To evaluate the predictive accuracy of the final multivariate model, in terms of discrimination, Area Under the ROC curve (AUC) was also reported. Moreover, to compare the actual vs. predicted by the model upstaging probabilities, a calibration plot was created. Patients were categorized in five classes, based on predicted probabilities (≤2%, 2–5%, 5–10%, 10–25% and >25%). For each class, the average of the predicted probabilities and the observed relative frequency of patients with upstaging were calculated and reported in the plot. 

All analyses were performed with SAS 9.4 (SAS Institute, Cary, NC, USA) and R version 4.0.3. 

## 3. Results

Among the 3100 VABBs analysed, 295 were diagnosed as low-grade DCIS and all the patients underwent subsequent surgical excision.

The clinicopathological features of the patients are summarized in Table 2.

Of these 295 patients, 272 were diagnosed by stereotactic VABB and identified by mammography (showing only microcalcifications), while 23 cases were diagnosed by ultrasound-guided VABB (showing as nodule). 

Such disproportion is due to the usual radiological manifestation of DCIS as microcalcifications, instead of nodules [20].

At the biopsy, the median age of patients was 51 (34–79) years, the median size of the lesion was 15 mm (4–100); radiological diagnoses were: 3 BIRADS 3 (1%); 124 BIRADS 4a (42%); 95 BIRADS 4b (32.2%); 61 BIRADS 4c (20.7%); 12 BIRADS 5 (4.1%).

In 128 (43.4%) of cases, the lesion was macroscopically removed by VABB. In 138 cases (51.1%) we identified the disease only in the cores with macrocalcifications.

The histological exam of the surgical specimens of the 295 patients indicated that 32 cases (10.8%) were upgraded to IDC, and 53 cases (18.0%) were upgraded to worst grade DCIS, intermediate grade DCIS with Ki-67 > 14% and, high-grade ductal carcinoma in situ.

Interestingly, in 61 cases (20.7%) only benign findings were observed in subsequent surgical specimens; in some cases, the VABB seems to be able to completely remove the lesion.

At univariate analysis, the upgrade rate to IDC was statistically lower in patients with greater age (*p* = 0.018), without post-biopsy residual lesion (*p* < 0.001), with a smaller post-biopsy residual lesion size (*p* < 0.001), and in presence of low-grade DCIS only in specimens with microcalcifications (*p* = 0.002). At multivariate analysis, only post-biopsy residual lesion (OR (95% CI) for patients with vs. without residual disease was 7.14 (1.58–32.2)) and disease only in cores with microcalcifications (OR (95% CI) for patients with vs. without disease with microcalcifications was 0.33 (0.13–0.83)) were significantly associated with the upstage at surgery (Table 3).

A nomogram for predicting the upstage at surgery, based on the multivariate logistic regression model, was reported in Figure 3. 

For example, a patient with: age at Mammotome biopsy = “67”, biopsy needle = “11G or 10G”, post biopsy residual disease = “No”, log_2_ (Post biopsy residual lesion size) = “3.32”, number of cores = “0”, disease only in cores with microcalcifications = “Yes”, has a probability of upstage (invasive at surgery) of 0.01. While a patient with: age at Mammotome biopsy = “62”, biopsy needle = “8G or 7G”, post biopsy residual disease = “Yes”, log_2_ (Post biopsy residual lesion size) = “5.91”, number of cores = “8”, disease only in cores with microcalcifications = “No”, has a probability of upstage (invasive at surgery) of 0.20.

An online Shiny application was developed for users to easily access the model (https://bagnardi.shinyapps.io/DCIS_upstage/, accessed on 7 January 2022).

The AUC of the final multivariate model was 0.795 (Figure 4, Panel A). Panel B of Figure 4 showed the calibration plot: as an example, the predicted probability of upstage at surgery was lower than 2% in 58 patients (average predicted risk in this category: 1.3%) and, among these 58 patients, only one (1.7%) upstage was observed; the predicted risk was greater than 25% in 46 patients (average predicted risk in this category: 31%) and, among these 46 patients, 17 (37%) upstages were observed.

The results of the univariate and multivariate analyses considering upstage to worst grade ductal carcinoma in situ at surgery (intermediate grade DCIS with ki-67 > 14% and high-grade DCIS) as dependent variable, are shown in Appendix A.

## 4. Discussion

DCIS is a non-life threatening condition and includes about 25% of all breast cancer cases. Most cases of DCIS will never progress to invasive breast cancer during a patient’s lifetime and the 20-year breast cancer-specific mortality rate in patients with DCIS is low [21,22,23]. 

Sagara and colleagues [7], in a recent publication, analysed surveillance, epidemiology, and end-results (SEER) data from nine US states involving 57,222 women with a median 72 months’ follow-up from diagnosis: the vast majority of patients diagnosed with all grades of DCIS (who did not receive surgery) did not decease from breast cancer. Considering this low long-term mortality, the surgical therapy and the radiotherapy of DCIS may be considered a sort of overtreatment and an unjustified cost to public health, especially for low-grade carcinomas in situ [24]. 

Four prospective international study protocols (LORIS, COMET, LORD, and LORETTA) are currently in place to evaluate non-invasive treatment strategies for DCIS the results of which will still be evaluated. However, the role of diagnostic underestimation of the breast biopsy is often overlooked. In a meta-analysis, Brennan et al. showed that 25.9% (18.6–37.2%) of presurgical cases diagnosed as DCIS were upgraded to IDC upon excision [8]. Considering only those undergoing VABB, this percentage dropped to around the 15% (regardless of the degree of DCIS) and to the 10% for the low-grade DCIS [25,26]. This percentage is still too high to propose active surveillance to a patient, as follow-up over surgery should be justified by an upgrade rate lower than 2%, as established for Breast Imaging Reporting and Data System, in which a possible diagnostic delay does not affect the outcome [15]. 

In our study, we propose a predictive model in order to minimize the risk of diagnostic underestimation in a smaller group of patients. Nomograms are predictive tools that allow, considering the multiples features, an assessment of the risk of underestimation [27]. With our nomogram, you are able to evaluate if a patient has a predicted probability of diagnostic underestimation below 2%. Notably, in almost 20% of those who underwent surgery, no residual disease was found in the surgical sample, suggesting a possible complete lesion removal by the VABB.

We believe that our predictive model, once validated in an external cohort, could help in the careful selection of patients to candidate to active surveillance rather than surgical excision. Our study may pose the basis for further future prospective studies where active surveillance can be suggested considering specific radiological and pathological criteria. 

The major limitation of our study is represented by its monocentric and retrospective nature, by the low number of cases considered, and by the lack of an external validation cohort. Our study has an exploratory nature: a step towards a long path that can avoid overtreatment in this category of patients.

To be concretely used in clinical practice, our model needs a rigorous validation in an external cohort and to be applied in a large number of patients.

However, we are convinced that these preliminary results are promising, easy to apply in all breast units and deserve to be further investigated in other studies.

We believe that the near future will increasingly focus on enhancing studies that allow us to identify patients in whom the risk of upstaging is lower. In this regard, studies aimed at verifying a specific gene expression of high-risk patients and aimed at verifying specific image features of the lesion with radiomics and contrast enhanced spectral mammography (CESM) will certainly have a fundamental role in this issue [28,29,30].

## 5. Conclusions

An easy-to-use predictive model that considers the size of the lesion, its complete removal with VABB, patient’s age, biopsy needle, number of cores and the presence of disease only in cores with microcalcifications is able to identify a population of patients with DCIS with low risk of upstaging to IDC.

These criteria, after validation in an external cohort, should be considered when selecting patients for active surveillance rather than surgical intervention.

## Figures and Tables

**Figure 1 cancers-14-00370-f001:**
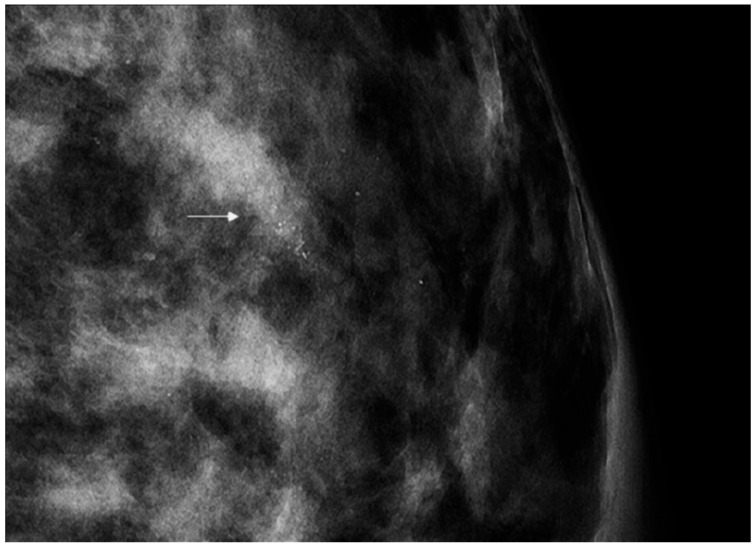
Full-field digital mammography showing a small cluster of pleomorphic microcalcifications (arrow) with a biopsy-proven histopathological result of low-grade ductal carcinoma in situ.

**Figure 2 cancers-14-00370-f002:**
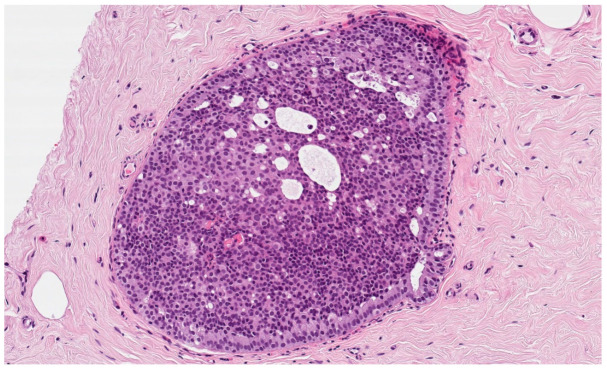
Histological features of low-grade DCIS from a breast biopsy showing bland homogeneous cells contained within the duct, forming rigid cell ‘bridges’ across the duct space in a cribriform architecture. In this case, the abnormal duct is surrounded by fibrotic stroma (hematoxylin and eosin, original magnification 100×).

**Figure 3 cancers-14-00370-f003:**
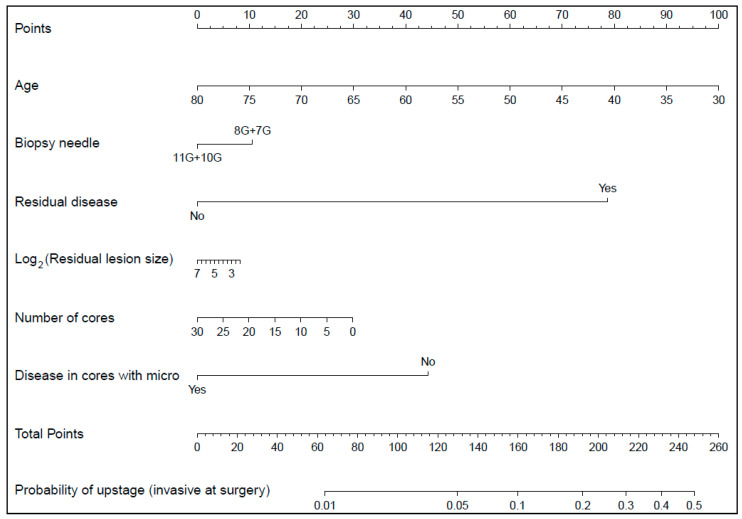
Nomogram for predicting the upstage (invasive at surgery) according to the multivariate logistic regression model. Instructions: to estimate the probability of upstage (invasive at surgery), locate the patient’s age at Mammotome biopsy on the “Age” axis. Draw a line straight upward to the point axis to determine how many points they receives for their age. Repeat the process for each additional variable. Sum the points for each of the predictors. Locate the final sum on the “Total point” axis. Draw a line straight down to find the patient’s probability of upstage (invasive at surgery). An online Shiny application was developed for users to easily access the model (https://bagnardi.shinyapps.io/DCIS_upstage/, accessed on 7 January 2022).

**Figure 4 cancers-14-00370-f004:**
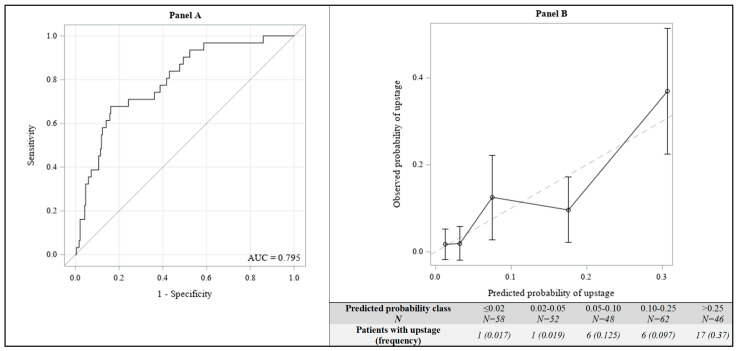
Predictive accuracy of the multivariate logistic regression model; ROC curve (**Panel A**) and Calibration plot (**Panel B**).

**Table 1 cancers-14-00370-t001:** Main features of the four prospective international study protocols (LORIS, COMET, LORD and LORETTA).

Study	LORIS [11]	COMET [12]	LORD [13]	LORETTA [14]
Country	UK	USA	EU	JAPAN
Year of activation	2014	2017	2017	2017
Accrual target (number of patients)	932	1200	1240	340
Size of the lesion	Any	Any	Any	<2.5 cm
Type of guide for biopsy	Stereotactic (vacuum assisted)	Stereotactic (vacuum assisted)	Stereotactic (vacuum assisted)	Stereotactic and ultrasound (vacuum assisted)
Hormone receptor status	Any	Hr*-positive only	Any	Hr*-positive only
Endocrine therapy	Optional	Optional	Not allowed	Mandatory
Minimum age at diagnosis	48	40	45	40
Comedonecrosis	Excluded	Allowed	Excluded	Excluded

Hr*: Hestrogen receptor.

**Table 2 cancers-14-00370-t002:** Distribution of patients, diagnostic and tumor characteristics (*N* = 295 DCIS low grade).

Variable	Level	Overall (*N* = 295)
Year of Mammotome biopsy, *N* (%)	1999–2004	66 (22.4)
	2005–2009	65 (22.0)
	2010–2014	97 (32.9)
	2015–2018	67 (22.7)
Days between mammography and Mammotome biopsy, median (min–max)		33 (0–313)
Missing		16
Age at Mammotome biopsy, median (min–max)		51 (34–79)
Biopsy needle, *N* (%)	8G + 7G	45 (15.5)
	11G + 10G	245 (84.5)
	Missing	5
Post biopsy residual disease, *N* (%)	No	128 (43.4)
	Yes	167 (56.6)
Post biopsy residual lesion size (mm), median (min–max)BIRADS, *N* (%)		15 (4–100)
	3	3 (1.0)
	4a	124 (42.0)
	4b	95 (32.2)
	4c	61 (20.7)
	5	12 (4.1)
Number of cores, median (min–max)		13 (0–30)
Disease only in cores with microcalcifications, *N* (%)	No	132 (48.9)
	Yes	138 (51.1)
	Missing	25
Days between Mammotome biopsy and surgery, median (min–max)		51 (5–247)
Missing		3
Outcomes of the study		
Upstage (invasive at surgery), *N* (%)	No	263 (89.2)
	Yes	32 (10.8)
Upstage at surgery (implying change of therapy), *N* (%)	No	242 (82.0)
	Yes	53 (18.0)
Absence of disease at the surgery, *N* (%)	No	234 (79.3)
	Yes	61 (20.7)

**Table 3 cancers-14-00370-t003:** Association between variables and upstage (invasive at surgery). Results from univariate and multivariate logistic regression analyses.

Variable	Level	Upstage/Tot (%)	Univariate Analysis			Multivariate Analysis ^1^		
			OR	95% CI	*p*-Value	OR	95% CI	*p*-Value
Overall	-	32/295 (10.8)	-	-	-	-	-	-
Age at Mammotome biopsy	+1 year		0.94	0.90–0.99	0.018	0.95	0.90–1.00	0.068
Biopsy needle	8G + 7G	7/45 (15.6)	Ref.	-	-	Ref.	-	-
	11G + 10G	25/245 (10.2)	0.62	0.25–1.53	0.30	0.77	0.29–2.06	0.60
	Missing	0/5						
Post biopsy residual disease	No	3/128 (2.3)	Ref.	-	-	Ref.	-	-
	Yes	29/167 (17.4)	8.76	2.60–29.4	<0.001	7.14	1.58–32.2	0.011
Post biopsy residual lesion size	+1 × log_2_ (mm)		1.76	1.26–2.46	<0.001	0.96	0.58–1.58	0.87
Number of cores	+1		0.98	0.91–1.05	0.53	0.98	0.90–1.05	0.53
Disease only in cores with microcalcifications	No	24/132 (18.2)	Ref.	-	-	Ref.	-	-
	Yes	7/138 (5.1)	0.24	0.10-0.58	0.002	0.33	0.13-0.83	0.018
	Missing	1/25						

^1^ Twenty-nine patients with at least one missing value among independent variables were excluded from the model. Goodness of fit statistics: McFadden’s R-Square = 0.16; AIC = 174.6; −2 Log Likelihood = 160.6.

## Data Availability

The data presented in this study are available on request from the corresponding author. The data are not publicly available due to privacy concerns, in accordance with GDPR.

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
