# Peer review of "A Model to Predict Upstaging to Invasive Carcinoma in Patients Preoperatively Diagnosed with Low-Grade Ductal Carcinoma In Situ of the Breast"

_cancers, 2022, doi:10.3390/cancers14020370_

Round 1
Reviewer 1 Report
I am grateful for the authors for addressing the issue I mentioned in my previous comments. I am satisfied with most of the responses and actions taken. Please see some additional comments below that may improve the manuscript further.
- I appreciate that the authors have included the continuous variables as such in the model. I do not understand what do you mean by level = “+5 years” in table 3. Could this be clarified? Was age since “Age at Mammotome biopsy” divided by 5 in the model? Why is that?
- I guess the Nomogram can be useful if a person does not have internet access, but it is not very practical. You could create a simple Shiny or Java app that calculates this, or even a simple shared google spreadsheet with the right formulas would be more handy. A link to this could be included in the paper, so that people can make quick calculations without having to have this graph at hand and use a ruler, and they would not have to manually log transform residual lesion size either.
- The pronouns in the instructions for the Nomogram seem odd. You use he/his/him in the text, but I guess that it is rare that the procedure is conducted on a man rather than a woman. So she/her seems to be more appropriate here (or a neutral they/their).
- You wrote that “in our hypothesis the diameter of the needle, which could be related to the diagnostic performance and therefore to the upstaging, is not considered a secondary variable”. I still don’t understand, how could the needle size, which is based on some random factors (availability) and some more or less arbitrary decision by the physician, could be related to upstaging risk. You are basically saying that needle size is a measure of diagnostic effectiveness. But I cannot see how the diagnostic effectiveness would affect the upstaging of the disease. Could this be clarified? Could this be explained in more detail in the paper, so that the readers see the theoretical rationale for including this as a predictor in the model?
- The new supplementary table is referred to in the text as Table 1s, but it is labeled Table S1 in the supplement. Please correct this inconsistency.
- “To be concretely used in clinical practice, our model needs a rigorous validation in an external court and to be applied in a large number of patients.” – “court” should be “cohort”
Author Response
I am grateful for the authors for addressing the issue I mentioned in my previous comments. I am satisfied with most of the responses and actions taken. Please see some additional comments below that may improve the manuscript further.
- I appreciate that the authors have included the continuous variables as such in the model. I do not understand what do you mean by level = “+5 years” in table 3. Could this be clarified? Was age since “Age at Mammotome biopsy” divided by 5 in the model? Why is that?
Yes, age at biopsy was divided by 5 in the model. Since this stylistic choice may be misleading to a reader, to avoid confusion we have now reported in the table the OR relative to a one-year increase.
- I guess the Nomogram can be useful if a person does not have internet access, but it is not very practical. You could create a simple Shiny or Java app that calculates this, or even a simple shared google spreadsheet with the right formulas would be more handy. A link to this could be included in the paper so that people can make quick calculations without having to have this graph at hand and use a ruler, and they would not have to manually log transform residual lesion size either.
We have now implemented the nomogram in an online calculator, available here:
https://bagnardi.shinyapps.io/DCIS_upstage/.
- The pronouns in the instructions for the Nomogram seem odd. You use he/his/him in the text, but I guess that it is rare that the procedure is conducted on a man rather than a woman. So she/her seems to be more appropriate here (or a neutral they/their).
Thanks for the comment. We changed the text according to your suggestion.
- You wrote that “in our hypothesis, the diameter of the needle, which could be related to the diagnostic performance and therefore to the upstaging, is not considered a secondary variable”. I still don’t understand, how could the needle size, which is based on some random factors (availability) and some more or less arbitrary decisions by the physician, could be related to upstaging risk. You are basically saying that needle size is a measure of diagnostic effectiveness. But I cannot see how the diagnostic effectiveness would affect the upstaging of the disease. Could this be clarified?
Could this be explained in more detail in the paper, so that the readers see the theoretical rationale for including this as a predictor in the model?
With a larger diameter of the needle (considering the larger opening diameter of the sampling window) with an equal number of biopsy samples, more material is collected, compared to sampling performed with a smaller needle. This allows the pathologist to have more material to analyze and therefore can be associated with a lower diagnostic underestimation. However, it is not always possible to use larger needles, in particular, the choice is made based on the thickness of the breast to be biopsied.
We explained this rationale in the text.
- The new supplementary table is referred to in the text as Table 1s, but it is labeled Table S1 in the supplement. Please correct this inconsistency.
We are sorry for the typo. We Corrected it according to your suggestion.
- “To be concretely used in clinical practice, our model needs a rigorous validation in an external court and to be applied in a large number of patients.” – “court” should be “cohort”
We are sorry for the typo. We Corrected it according to your suggestion.
Reviewer 2 Report
I believe the manuscript has been significantly improved and now warrants publication in Cancers.
Author Response
I believe the manuscript has been significantly improved and now warrants publication in Cancers.
We want to thank the reviewer for the positive comment. We think that our work could be useful to deepen the knowledge on biopsy underestimation in carcinomas in situ of the breast.
This manuscript is a resubmission of an earlier submission. The following is a list of the peer review reports and author responses from that submission.
Round 1
Reviewer 1 Report
The manuscript describes a retrospective analysis of existing data on 295 cases diagnosed with low-grade DCIS, with the goal of building a model that can predict patients having a low (<2%) risk of upstaging. This is a very important goal that has potentially huge benefits for patients, preventing unnecessary overtreatment and invasive intervention. The present study is a minor step toward reaching this goal due to its limitations, but every step taken towards this goal may be important. On the other hand, I don’t feel that the authors correctly represent the huge limitations of this study, so I would like to encourage the authors to present these limitations very clearly in the manuscript so the readers understand that the model presented here is but an exploratory first step in a longer journey. Below I list detailed comments which might help the authors improve the manuscript:
- The last paragraph in the introduction section seems inappropriate/incomplete to me. You mention four ongoing studies and that “selection of patient population based on clinical and radiological features (which may reduce the diagnostic underestimation of the biopsy) appears neglected in these protocols”. Could be more specific about how this is relevant to the present paper? Also, I am wondering whether this is the right concluding paragraph to the intro. The concluding paragraph usually states the aims and or research questions, or summarizes the rationale for the study. The current final paragraph to the intro does none of this, or I don’t see how. Maybe this paragraph could be moved earlier in the intro?
- The last paragraph in the introduction referencing Table 1. mentions four international protocols: LORIS, COMET, LORD, and LORETTA, while Table 1 contains the headings: LORIS, COMET, LORD, and LORIS, so LORETTA does not appear in Table 1. Is this a typo?
- Some of the predictors included in the model are continuous variables that have been categorized (age, residual lesion size, number of cores). This leads to a loss in statistical power and imprecision the closer we are to the cutoff point of the categories. I think most of these predictors should be used as continuous variables in the prediction models instead of categorical predictors. If a predictor is used as a categorical variable instead of a contiuous variable, there needs to be a clear rationale for that stated in the paper for why do we think that these categories actually exist (instead of being just arbitrary cutoffs). There might be such rationale, for example I can see that age 50+ is related to some degree to menopause, but if the reason for the categorization is the hormone levels or other related features to menopause, why don’t you use these as predictors rather than age? It is not enough rationale that “it makes diagnostic decision easier with discrete categories”. The decision thresholds can be set up later with discrete cutoff points , but the model should still be trained with the most precision possible, and that requires that continuous predictors be kept continuous in the model.
- I don’t really understand why is needle size one of the predictors. What is the proposed mechanism involved here? Since I am not a physician I am not sure what is involved in deciding the needle size used for the biopsy, but I presume that this decision is based on the characteristics of the lesion. So why not use these characteristics, instead of the secondary variable which is derived from them, the size of the needle? Is there some reason to suspect that the size of the needle would somehow be causally related to upstaging?
- The conclusion states a different final model than the one listed in the results section: “An easy to use predictive model that considers the size of the lesion, its complete removal with VABB and the presence of disease only in cores with microcalcifications is able to identify a population of patients with DCIS with low risk of upstaging to IDC.” Age does not appear in the conclusion sentence, while age is in the final model.
- As I understand based on the manuscript, the authors used “invasive at the time of surgery” as the outcome variable in their model. But the authors seem to use this as a synonym of upstage risk or upstage rate. The variability of the days between mammotome biopsy and surgery seems to be pretty big in the sample, and on average the surgery seems to follow the biopsy within 2 months. But since upstaging could happen over time, the fact that upstaging did not happen at the time of surgery (which for some patients came just a few days after the biopsy) does not mean that it would not happen eventually. So I think it is important to keep this in mind and use the right language throughout the manuscript to avoid conflating upstage status at the time of surgery with upstage risk or upstage rate.
- Two possible criteria for upstaging are mentioned in the manuscript, yet, the statistical models are only reported with one of them (invasive at surgery) as an outcome. It should be calrified why do the authors focus on one over the other, and if possible, results of the models should be presented for the outher outcome as well (if this would take too much space in the main manuscript, this could be presented for the other outcome in an online supplement.
- The description of the prediction accuracy of the final logistic regression model is missing. Please, present the McFadden R^2, the AIC, and the -2ll of the model. Also, present the actual upstage at surgery rates next to the predicted probabilities. It seems that the goal of the authors is to be able to predict whether the upstage risk for a person is 2% or lower, in which case they would recommend active surveillance instead of surgery. So present the number and percentage of people for whom the model predicted 2% or lower upstage at surgery risk, and the actual percentage of upstage at surgery in this group of patients.
- The result-based model selection used by the authors poses a large risk for overfitting. It is important that the predictors in prediction models are based primarily on theory, and prior findings. The manuscript should give a justification (theoretical as well as based on prior findings) for the inclusion of each predictor in this analysis. The result-based model selection that is used by the authors here (the inclusion of only the significant predictors) should only be regarded as an exploratory tool, which is mostly used in practice when there are too many predictors to be practically or statistically useful. The number of predictors do not seem to be too much here, so I am not really sure what is the reason for using such risky statistical approach here. Why not just build a model including all of the predictors that are important based on theory and prior findings? This should be clearly explained in the manuscript or the pruning of variable should be abandoned for a more complete model.
- The authors are correct to say that the model needs to be validated in an external cohort, but the exploratory nature of the approach used in this study should be better emphasized in the manuscript. The risk of overfitting should be mentioned and explained in detail in the limitations section, and this should also be mentioned when describing the statistical analysis used. It should also be clearly emphasized how small the sample size is in this study compared to the ambitious goal of the model. The goal of the model on the long run is to be able to reliably identify people who have 2% or lower risk for upstaging. Lets say that the risk of upstaging is 1% in the subgroup of patients who have age of 50+, disease only in cores with microcalcifications, and with no post-biopsy residual lesion. To show this reliably, we would need more than a 1000 people just in this subgroup. How many such patients do the authors have in their current sample? Probably not a lot. This limitation should be clearly emphasized, not just mentioned in a middle of a short sentence as “relatively low number of cases considered”, like they do now.
- Please, share the statistical analysis code used for the study as a supplement.
- Please, share the data used in this study so that the readers can re-analyse and potentially improve on the models the authors built. If the data cannot be shared for some reason, the authors should clearly state the reason for this, and potential alternative ways the readers can get access to the data.
Reviewer 2 Report
Results are only presented by tables, perhaps some graphs would be more suitable.
Text is not justified. There is an extra space in line 58. There is an extra enter in reference 15. Title of table 2 should be in the page 6. An enter is needed before Statistical analysis (line 158).
I think it would be interesting if authors could explain in the conclusions section what are the future steps in their research.